# Large-Scale Synthesis of Semiconducting Cu(In,Ga)Se_2_ Nanoparticles for Screen Printing Application

**DOI:** 10.3390/nano11051148

**Published:** 2021-04-28

**Authors:** Bruna F. Gonçalves, Alec P. LaGrow, Sergey Pyrlin, Bryan Owens-Baird, Gabriela Botelho, Luis S. A. Marques, Marta M. D. Ramos, Kirill Kovnir, Senentxu Lanceros-Mendez, Yury V. Kolen’ko

**Affiliations:** 1International Iberian Nanotechnology Laboratory, 4715-330 Braga, Portugal; bruna.goncalves@inl.int (B.F.G.); alec.lagrow@inl.int (A.P.L.); 2Center of Physics, University of Minho, 4710-057 Braga, Portugal; pyrlinsv@fisica.uminho.pt (S.P.); lsam@fisica.uminho.pt (L.S.A.M.); marta@fisica.uminho.pt (M.M.D.R.); senentxu.lanceros@bcmaterials.net (S.L.-M.); 3Center of Chemistry, University of Minho, 4710-057 Braga, Portugal; gbotelho@quimica.uminho.pt; 4Department of Chemistry, Iowa State University, Ames, IA 50011, USA; bowens@iastate.edu (B.O.-B.); kovnir@iastate.edu (K.K.); 5Ames Laboratory, U.S. Department of Energy, Ames, IA 50011, USA; 6BCMaterials, Basque Center for Materials, Applications and Nanostructures, UPV/EHU Science Park, 48940 Leioa, Spain; 7Ikerbasque, Basque Foundation for Science, 48009 Bilbao, Spain

**Keywords:** Cu(In,Ga)Se_2_, nanoparticles, wurtzite-type, screen printing photoabsorber

## Abstract

During the last few decades, the interest over chalcopyrite and related photovoltaics has been growing due the outstanding structural and electrical properties of the thin-film Cu(In,Ga)Se_2_ photoabsorber. More recently, thin film deposition through solution processing has gained increasing attention from the industry, due to the potential low-cost and high-throughput production. To this end, the elimination of the selenization procedure in the synthesis of Cu(In,Ga)Se_2_ nanoparticles with following dispersion into ink formulations for printing/coating deposition processes are of high relevance. However, most of the reported syntheses procedures give access to tetragonal chalcopyrite Cu(In,Ga)Se_2_ nanoparticles, whereas methods to obtain other structures are scarce. Herein, we report a large-scale synthesis of high-quality Cu(In,Ga)Se_2_ nanoparticles with wurtzite hexagonal structure, with sizes of 10–70 nm, wide absorption in visible to near-infrared regions, and [Cu]/[In + Ga] ≈ 0.8 and [Ga]/[Ga + In] ≈ 0.3 metal ratios. The inclusion of the synthesized NPs into a water-based ink formulation for screen printing deposition results in thin films with homogenous thickness of ≈4.5 µm, paving the way towards environmentally friendly roll-to-roll production of photovoltaic systems.

## 1. Introduction

Over the last few decades, Cu(In,Ga)Se_2_ (CIGS) compound has been attracting considerable attention for the fabrication of second-generation photovoltaics (PVs). This *p*-type semiconductor exhibits high absorption coefficient of ≈10^5^ cm^−1^ and chemically tunable direct band gap between 1.0 and 1.7 eV [1]. Remarkably, the champion laboratory-produced CIGS PV device has reached 23.4% of efficiency using vacuum deposition processes [2]. Solution-processed CIGS PVs give access to semi-transparent, light weight, and flexible PV devices [3], opening widely their range of application from windows [4] to space exploration [5]. Moreover, solution processing methodologies are compatible with roll-to-roll production of PV devices, rendering solution processing more cost efficient than vacuum deposition methodologies. To date, a maximum efficiency of 17.3% [6] has been achieved using solution processing methodologies for CIGS PVs. Recently, we embarked on the development of sustainable methodologies for the fabrication of CIGS PVs, resulting in a device with over 6% of efficiency [7]. This fabrication relies on the formulation of an ink based on CuO, In_2_O_3_, and Ga_2_O_3_ nanoparticles (NPs), and subsequent screen printing of the ink followed by selenization [8], which converts the resultant thin film of oxide NPs into the desired CIGS phase. An interesting feature of such printing fabrication processes is that it is more sustainable in comparison to classical CIGS PV fabrication based on energy-demanding vacuum methods [9]. Nevertheless, the employment of a selenization step during the fabrication of the CIGS PV is a shortcoming due to the potential risk it carries through the high toxicity of the evolved selenium species, i.e., Se vapor and H_2_Se gas.

To overcome this shortcoming, fabrication of the CIGS PVs directly from CIGS NPs would eliminate the need for the selenization step, thus lowering the environmental impact of the solar cell production. In this context, the synthesis of CIGS NPs has attracted considerable interest, and reports based on hot-injection [10,11], heat-up [12,13], solvothermal [14,15], hydrothermal [16,17], or mechanochemical [18,19] methods have appeared, usually delivering crystalline and phase-pure CIGS NPs [20]. Notably, most of the synthesis protocols, regardless of the method employed [21,22], are severely limited to the preparation of CIGS NPs with common tetragonal chalcopyrite-type structure (space group $I\bar{4}2d$), the most thermodynamically stable phase at room temperature [21,22]. There are only a few reports on the synthesis of CuInSe_2_ (CIS) NPs with uncommon hexagonal wurtzite-type structure (space group *P*6_3_*mc*) [21,23,24,25,26], mostly due to the challenging control over stoichiometry and crystal structure [27]. To the best of our knowledge, only one synthesis strategy has given access to quaternary CuIn_1−*x*_Ga*_x_*Se_2_ NPs [28]. Most likely, this could be associated with the addition of Ga into the system, which significantly slows the nucleation and growth kinetics of the NPs, turning difficult its incorporation into CIS [27]. On the other hand, the presence of Ga in the CIGS photoabsorber is essential, since the stoichiometry [Cu]/[In + Ga] = 0.8–0.9 and [Ga]/[In + Ga] = 0.3 leads to highly efficient PV cells [29,30].

Very recently, we reported a straightforward heating-up synthesis of ternary wurtzite-type CIS NPs [25]. Moreover, we discovered an elegant chemical ordering in the as-synthesized NPs, wherein Cu and In segregate over distinct framework positions within the perfect Se wurtzite sublattice resulting in hexagonal plate-like morphology. Herein, we extend our synthesis methodology towards the preparation of quaternary CIGS NPs with uncommon wurtzite-type structure, along with upscaling of the procedure to 5 g scale. The experimental and theoretical results on Ga incorporation into the Cu–In–Se system are discussed, while the thermal stability of the resultant NPs is investigated by in-situ powder X-ray diffraction and in situ electron microscopy. We further show that we can formulate an eco-friendly ink based on the synthesized CIGS NPs to screen-print phase-pure semiconducting thin films.

## 2. Experimental

### 2.1. General Reagent Information

Tetrakis(acetonitrile)copper(I) tetrafluoroborate (TACT, 98%, TCI, Oxford, UK), indium(III) acetate (In(ac)_3_, 99.99%, Sigma-Aldrich, St. Louis, MO, USA), gallium(III) acetylacetonate (Ga(aca)_3_, 99.99%, Sigma-Aldrich, St. Louis, MO, USA), hexadecylamine (HDA, 95%, TCI, Oxford, UK, melting point 44 °C, boiling point 330 °C), diphenyl diselenide (Ph_2_Se_2_, 97%, TCI, Oxford, UK), acetonitrile (ACN, 99.9%, Fisher Scientific, Waltham, MA, USA), ethylenediamine (EDA, ≥99%, Sigma-Aldrich, St. Louis, MO, USA), dichloromethane (DCM, ≥99.8%, Fisher Scientific, Waltham, MA, USA), hydroxypropyl methyl cellulose polymer (HPMC, 2% aqueous solution, viscosity 80–120 cP, Sigma-Aldrich, St. Louis, MO, USA), toluene (≥99.5%, Sigma-Aldrich, St. Louis, MO, USA), ethanol (≥99.8%, Honeywell, Charlotte, NC, USA), acetone (≥99.5%, Honeywell, Charlotte, NC, USA), and isopropanol (≥99.8%, Honeywell, Charlotte, NC, USA) were used as received. Ultrapure water (18.2 MΩ cm) was generated using Milli-Q Advantage A10 system (Millipore, Burlington, MA, USA).

### 2.2. Synthesis

The synthesis was carried out using standard Schlenk line techniques. Initially, the reaction with the following metals ratio [Cu]/[In + Ga] = 0.83 and [Ga]/[In + Ga] = 0.3 was used. For this purpose, TACT (12.40 mmol), In(ac)_3_ (10.8 mmol), Ga(acac)_3_ (4.1 mmol), Ph_2_Se_2_ (20.2 mmol), and HDA (414.10 mmol) were charged into a 500 mL three-neck round-bottom flask. The flask was connected to a condenser and equipped with a magnetic stirrer, thermocouple, and vacuum adapter. After attaching the flask to a Schlenk line under Ar, the system was slowly heated to 90 °C and the precursors were dissolved in melted HDA under stirring. After complete dissolution, as observed by the emergence of a clear green solution, the reaction was degassed under vacuum for 30 min to remove undesired low boiling point liquids, as possible water and acetic acid admixtures. The reaction mixture was placed under Ar and the system was rapidly heated to 300 °C and stirred at this temperature for 1 h. As the reaction proceeds, the formation of a brown-black slurry was observed. The slurry was cooled to 70 °C and diluted with 100 mL of toluene, followed by cooling to room temperature (RT). Notably, it is important to conduct the dilution at 70 °C, since HDA is solid at RT, and therefore, it will be difficult to isolate the product without dilution. Next, the resultant NPs were precipitated by a mixture of toluene/ethanol (3:1), washed with the same solvent, and collected by centrifugation (9000 rpm, 5 min). The washing procedure was repeated three times in total. Finally, the NPs were dried under vacuum overnight and homogenized using an agate mortar, thus affording ≈5 g of the product as powder (**Sample I**).

Similarly, **Sample II** was synthesized by modifying the initial concentrations of the metal precursors to give access to CIGS NPs with the desired metal ratio. Specifically, TACT (8.20 mmol), In(ac)_3_ (6.43 mmol), Ga(acac)_3_ (2.65 mmol), and same amount of Ph_2_Se_2_ and HDA as used in the previous synthesis, were used during the synthesis, thus affording ca. 5 g of **Sample II**.

### 2.3. Ligand Exchange

For ink formulation purposes, a ligand exchange procedure was employed to replace HDA, since it is solid at RT. To this end, 120 mL of ACN, 1 mL of EDA, and 15 mL of DCM were loaded into a flask and stirred magnetically at RT [31]. After a few minutes, 3 g of the synthesized NPs was added, and the solution was stirred at RT for 24 h. The NPs were then collected by centrifugation (9000 rpm, 10 min) and dried under vacuum. The resultant powder was subjected to wet ball milling (WBM) to eliminate possible agglomerates and improve the dispersion of the synthesized NPs in the ink. To this end, a dispersion of 3 g of NPs in 5 mL of IPA was ball-milled for 24 h using a KD-6808 rotary polishing machine (Guangzhou 7 Gram Machinery Equipment Co., Ltd., Guangzhou, China) with bidirectional rotation and YSZ balls with two sizes of 0.2 and 10 mm (mass ratio 50:50). Finally, the resultant suspension was filtered using a 1 µm syringe filter to homogenize the NP size and dried at 80 °C. The resultant powder was ground in a mortar and preserved for ink formulation.

### 2.4. Ink Formulation

Based on our previous report [32], the water-based ink was formulated in a 10 mL glass vial. HPMC (45 mg) was first dissolved in a mixture of water (0.33 mL) and ethanol (0.66 mL) by magnetic stirring for 4 h at RT. Then, the NP powder (0.60 g) was added to the resulting viscous 5% HPMC solution and kept under stirring for 12 h, resulting in a homogenous water-based ink with 40% NP content.

### 2.5. Screen Printing

For printing, 2.6 × 2.6 cm^2^ soda-lime glass (SLG, Fisher Scientific, Waltham, MA, USA) substrates with 1 mm of thickness were consecutively cleaned in acetone, IPA, and water using ultrasonication (Elmasonic P30H, Elma, Singen, Germany) at 60 °C for 20 min each. The substrates were then rinsed with ethanol, dried under N_2_ flow, and subjected to O_2_ plasma treatment (Harrick Plasma, Ithaca, NY, USA) during 10 min for a complete surface cleaning.

Square patterns of 2.5 × 2.5 cm^2^ were printed above the previously cleaned SLG substrates using a semi-automatic screen printer (DX-3050D, Shenzhen Dstar Machine Co., Ltd., Shenzhen, China) equipped with a vacuum stage to hold the substrate. Thin films were screen-printed using 180 threads cm^−1^ count with thread diameter of 27 µm and mesh opening of 24 µm. Three-step printing was employed followed by immediate drying at 90 °C to evaporate the solvent, followed by annealing to eliminate organic matter inside a quartz tubular furnace (Termolab, Aveiro, Portugal) using 100 sccm of Ar at 500 °C with a heating rate of 50 °C min during 50 min in total. Above the same thin film, an additional two-step printing was employed to fill the voids left by the previous printing, followed by the described thermal treatments. The printing process was performed using an 85-shore squeegee at 0.3 ms^−1^ of velocity with a 75° deflection angle and a distance between the mesh and the substrate of 5 mm.

### 2.6. Characterization

**Powder X-ray diffraction (XRD).** The phase composition of the NP powder was evaluated on an X’Pert PRO diffractometer (PANalytical, Malvern, UK) set at 45 kV and 40 mA, equipped with a Ni-filtered Cu Kα radiation and PIXcel detector. XRD data were collected using Bragg–Brentano geometry in a 2*θ* range from 15° to 80° with a scan speed of 0.01° s^−1^. The XRD patterns were matched to the International Centre for Diffraction Data (ICDD) PDF-4 database using HighScore software package (PANalytical, Malvern, UK).

The average crystallite size was estimated in HighScore software using the Scherrer equation: $D=\frac{K\lambda }{\beta cos\theta }$, where *D* is the crystallite size, *K* is the Scherrer constant (0.89), *λ* is the X-ray wavelength, *β* is the width of the peak (full width at half the maximum (FWHM) in radians), and *θ* is the Bragg angle.

**Variable Temperature In-situ Synchrotron Powder XRD.** Variable temperature in-situ powder XRD data was collected at the synchrotron beamline 17-BM at the Advanced Photon Source, Argonne National Laboratory. Wurtzite CIS sample was loaded into 0.5/0.7 mm inner/outer diameter silica capillaries and sealed under vacuum. The sealed silica capillaries were placed into a secondary shield capillary, with a thermocouple set as close as possible to the measurement area. Details for experimental set-up are provide in [33]. The data were collected with *λ* = 0.24141 Å at variable temperatures.

**Raman spectroscopy.** Local phase composition of the NP powders was analyzed by an alpha300 R confocal Raman microscope (WITec, Ulm, Germany) with a 532 nm Nd:YAG laser using 0.9 mW of power focused on the specimen with a ×50 lens (Zeiss, Oberkochen, Germany). Raman spectra were collected using 1800 groove mm^−1^ grating with 100 acquisitions and 1.5 s of acquisition time.

**Electron microscopy.** The evaluation of the surface and cross-sectional morphologies of the printed thin films as well as the chemical composition of the NP powders were performed by scanning electron microscopy (SEM) using a Quanta 650 FEG ESEM microscope (FEI co., Hillsboro, OR, USA) equipped with energy-dispersive X-ray spectroscopy (EDX, FEI co., Hillsboro, OR, USA). To investigate fine microstructure and the chemical composition of the synthesized NPs, high-angle annular dark-field scanning transmission electron microscopy (HAADF–STEM), selected area electron diffraction (SAED), and energy-dispersive X-ray spectroscopy in STEM mode (STEM–EDX) were performed using a Titan Themis Titan Themis 60–300 (FEI co., Hillsboro, OR, USA) equipped with an X-FEG gun, superX EDX configuration with four detector system, an image corrector, and probe corrector, operating at 200 kV.

The in-situ heating studies were carried out on the Titan Themis (FEI co., Hillsboro, OR, USA) with the NanoEx i/v heating holder, with microelectromechanical systems (MEMS, Thermo Fisher Scientific, Waltham, MA, USA) chips.

**Optical properties.** The optical band gap measurements on the as-synthesized NP powders were performed using UV−Vis−NIR spectroscopy. The resulting data were collected at RT using a LAMBDA 950 UV/Vis/NIR spectrophotometer (PerkinElmer, Waltham, MA, USA) equipped with a 60 mm integrating sphere and InGaAs detector.

**Surface tension and rheological properties**. The surface tension of the ink was measured by drop-shape analysis-contact angle method (KRÜSS, Hamburg, Germany) with DSA3 software package (KRÜSS, Hamburg, Germany), using a drop volume of 10 µL and a needle with 0.9 mm diameter, at RT. The dynamic viscosity measurements were conducted at RT on a MCR 300 modular compact rheometer (Physica, Anton Paar GmbH, Graz, Austria) using a shear rate in the range of 0–500 s^−1^.

**Thermogravimetric analysis (TGA).** The thermal behavior of the formulated ink was obtained with a TGA/DSC 1 STARe system (Mettler Toledo, Columbus, OH, USA) under Ar flow with a heating ramp of 10 °C min^−1^.

**In silico study.** Density functional theory (DFT) calculations of segregation energy where conducted using VASP DFT package (ver. 5.3.3, VASP Software GmbH, Vienna, Austria) for high accuracy [34]. For simulations of the effect of passivating ligands, another DFT package—SIESTA (ver. psml-R1, open source community project) [35]—was used as a trade of between accuracy and computational efficiency for molecular systems. A 128-atom 2 × 2 × 2 supercell of a hexagonal CIS lattice was used both for bulk and surface calculations with 1.4 nm of vacuum layer added in the latter case. Reciprocal space was sampled using 2 × 2 × 2 Monkhorst-Pack grid. In both cases, Perdew-Burke-Ernzerhof generalized gradient approximation [36] was used for exchange and correlation functional during the geometry relaxation and the total energies and forces were converged down to 10^−4^ and 0.05 eV Ang^−1^ per atom. Energy cutoffs of 520 eV and 800 Ry were used correspondingly for VASP and SIESTA calculations. Scalar-relativistic PSML [37] pseudopotentials were used for the latter. To establish the preferred atomic arrangement on the slab surface where the candidate structures had different atomic composition (Cu- or In-rich), formation enthalpy difference for super-cells with different atomic composition was computed as $\Delta H_{form}=\Delta E_{DFT}-\sum \Delta n_{i}\mu_{i}$, where the first term is the total energy difference between candidate structures per super-cell and $\mu_{i}=E_{i,bulk}/N_{i,bulk}$—chemical potential of element *i* (Cu or In), estimated from the energy per atom of a corresponding single element crystal.

## 3. Results

We previously synthesized CIS NPs [25] with excellent crystallinity and phase-pure hexagonal wurtzite structure. In this study, we sought to extend the scope to CIGS NPs as well as investigating their applicability for screen printing deposition, which is relevant for roll-to-roll production of PVs. The large-scale synthesis was conducted by reacting Cu^+^, In^3+^, and Ga^3+^ precursors with Ph_2_Se_2_ in HDA working both as solvent, due to its high boiling point, and as capping agent. First, a synthesis with [Cu]/[In + Ga] = 0.83 and [Ga]/[In + Ga] = 0.3 nominal stoichiometry of the metal precursors was conducted, however, the resultant NPs metal ratio was different from the nominal (**Sample I**) (*vide infra*). Therefore, the ratio of the starting materials was readjusted, which resulted in NPs with the targeted composition (**Sample II**). The conducted syntheses delivered ~4.5 g (~90% yield) and ~3.5 g (~70% yield) of CIGS NPs for **Samples I** and **II**, respectively.

The XRD phase composition analysis of the resulting NPs from both syntheses (Figure 1a) revealed major peaks corresponding to (100), (002), (101), (102), and (110) reflections of wurtzite with relative intensities and positions matching well with the reported characteristic peaks of wurtzite CIS [38]. Notably, no signs of chalcopyrite phase or any secondary phases were found, revealing phase-pure CIGS NPs with rare wurtzite-type hexagonal structure (space group *P*6_3_*mc*). The Raman data (Figure 1b) revealed a sharp peak at around 178 cm^−1^ for **Sample I**, corresponding to the A_1_ vibrational mode of CIS [25]. On the other hand, the spectrum of **Sample II** confirmed the presence of phase-pure CIGS NPs, showing a sharp peak at around 174 cm^−1^, corresponding to A_1_ vibrational mode of CIGS and two broad bands at 127 and 211 cm^−1^ corresponding to the B_1_ and B_2_/E modes, respectively [39,40]. Importantly, both spectra were found to be free of CuSe_2_ secondary phase, which usually emerges as an additional Raman peak at 260 cm^−1^, and is known to be a detrimental phase for PVs by functioning as a recombination center for charge carriers through the photoabsorber. The chemical composition of both NP samples was studied by SEM–EDX (Appendix A), which revealed a nominal metal ratio of Cu_1.13_(In_1.42_Ga_0.20_)Se_2_ for **Sample I** and the successful readjustment for **Sample II** to Cu_0.89_(In_0.72_Ga_0.28_)Se_2_, well matching with the targeted composition [Cu]/[In + Ga] ≈ 0.8 and [Ga]/[Ga + In] ≈ 0.3. The crystallite size of NPs from **Sample II** was estimated using Scherrer formula, revealing an average of 29 ± 8 nm.

The HAADF–STEM images with STEM–EDX mapping (Appendix A) of **Sample I** confirmed the presence of hexagonal plates with 10–80 nm size. Interestingly, although the maps revealed a uniform distribution of Cu, In, and Se metals, Ga seemed to be segregated at the surface of the NPs. The observed Ga segregation and the Raman spectrum from **Sample I** has driven us to assume to be on the presence of (CIS + Ga phases). On the other hand, hexagonal plates with a size of 10–70 nm are observed in the HAADF–STEM images of **Sample II** along the (001) zone axis (Figure 2a,c). Moreover, the fast Fourier-transform (FFT) patterns (Figure 2b) confirm the hexagonal wurtzite phase of the NPs with superstructural ordering with twin planes in the superstructure ordering shown by the streaking in the FFT (Figure 2b) and by the red arrows and dashed lines (Figure 2c). In the STEM–EDX maps (Figure 2d), a more uniform distribution of all metals is detected as compared to **Sample I**, with Ga located not only on the surface but also inside the NPs, revealing pure-phase CIGS with hexagonal wurtzite structure. Therefore, we selected **Sample II** to move forward with the study.

The deposition of a photoabsorber layer requires an annealing treatment to remove organic matter from the HDA stabilizer, which would hamper the performance of the PV device since such impurities will affect the electrical properties of the films. Thus, the thermal stability of the synthesized NPs was evaluated by TGA (Appendix A), which revealed that at 500 °C, all organic matter is degraded, with a weight loss of ~14.5%, leading us to select this temperature for the annealing.

Interestingly, our previously reported synthesis [25] revealed that the synthesized CIS NPs tend to have superstructural ordering. Wurtzite is considered a metastable phase of CIS that is difficult to stabilize in comparison to the thermodynamically stable chalcopyrite CIS. Thus, prior to annealing the wurtzite films, we performed thermal stability studies on the CIS NPs by in-situ XRD (Figure 3), where the wurtzite phase was found to be stable up to 400 °C. At higher temperatures, wurtzite CIS started to transform into the thermodynamically stable chalcopyrite phase. No further changes were observed in 480–540 °C range. Due the similarities between the CIS and CIGS NPs, we assumed similar thermal behavior for the herein synthesized CIGS NPs. To complete the thermal stability studies, in-situ TEM imaging was performed under vacuum on the synthesized CIGS NPs (Figure 4). Similar to previously reported CIS NPs, the as-synthesized CIGS NPs present hexagonal wurtzite-type structure (*P*6_3_*mc* subcell) with superstructural ordering. The NPs were heated rapidly to 450 °C over the course of ~10 min and then left at 450 °C. The conducted experiment revealed no changes in the crystal lattice bellow 450 °C, however, after ~25 min at 450 °C, the supercell lattices of the wurtzite phase disappeared, as indicated by the calculated FFT patterns.

With these high-quality NPs in hand, we proceeded for the development of a water-based ink for the screen-printed thin film to be employed as photoabsorber layer. To this end, a ligand exchange procedure was successfully employed, as detected by the NPs TGA curve (Appendix A), to replace HDA, which is a solid at RT, by EDA. The optical band gap of the NPs after ligand exchange was evaluated through the absorption spectrum resulting from UV–Vis–NIR measurements (Appendix A), which revealed a strong absorption of light from the entire visible to near-infrared regions. Notably, no significant differences on the optical properties were found between the HDA and EDA-capped NPs. The band gap of EDA-capped NPs ($E_{g}$) was determined to be 0.95(2) eV, which is in good agreement with reported bandgaps for wurtzite CIS NPs [25]. The calculations using Tauc plot, (*Ahν*)^2^ vs. energy, resulted in a similar band gap of 0.98(4) eV.

The conducted ligand exchange procedure allowed for the dispersion of the NPs in water/ethanol solvents comprising HPMC as a rheological additive for ink formulation, based on [32]. Notably, the use of WBM procedure gave access to a smoother screen-printed film with improved NPs dispersion (Appendix A). The resultant ink, with 40% NP content, revealed a non-Newtonian behavior, with a range of dynamic viscosity of 1.3–4.0 Pa s (Appendix A) and a surface tension of 34.3 mN m^−1^, suitable for screen printing deposition over SLG. Moreover, the TGA of the ink (Appendix A) revealed that 500 °C should be employed to remove organic matter from EDA and HPMC. After employing five-step screen printing of the developed ink and annealing the resultant film at 500 °C, a porous photoabsorber layer with homogenous thickness of ≈4.5 µm was observed through SEM surface and cross-sectional imaging (Figure 5a,b). HAADF–STEM data (Figure 5c,d) showed CIGS NPs surrounded by a tiny layer of carbon, probably arising from the rheological HPMC additive used in the ink formulation, suggesting that a longer annealing treatment should be performed. STEM–EDX (Figure 5e) reveals a more uniform distribution of the metals as compared to the powder STEM–EDX analysis.

The resultant screen-printed film after annealing is a phase-pure CIGS with tetragonal chalcopyrite structure (space group $I\bar{4}2d$) with no signs of undesirable secondary phases (Appendix A). The major peaks found at 26.8°, 44.6°, and 52.9° correspond to (112), (220/204), and (312/116) reflections of chalcopyrite CIGS phase [14,41], respectively. The Raman data (Appendix A) revealed a sharp peak at around 174 cm^−1^, corresponding to the A_1_ vibrational mode of CIGS and two broad bands at 126 and 209 cm^−1^ corresponding to the B_1_ and B_2_/E modes, respectively [39,40], without presence of CuSe_2_ secondary phase.

## 4. Discussion

Based on our previous success in the synthesis of high-quality CIS NPs with the hexagonal wurtzite structure [25], we adapted this procedure for the synthesis of CIGS NPs synthesis. The developed synthesis delivered ~3.5 g of high-quality phase-pure CIGS NPs with hexagonal wurtzite phase. Although several strategies have been reported for the synthesis of tetragonal chalcopyrite CIGS NPs, syntheses of hexagonal wurtzite CIGS is uncommon. Importantly, wurtzite metastable phase is characterized by an increased amount of bonds between Cu and Se atoms, resulting in an improved band structure in regard to the electron transition and transport due to the delocalized d electrons of Cu [42]. Accordingly, the absorption of light from visible and infrared regions is higher for wurtzite phase than for the chalcopyrite one, which can result in improved photovoltaic efficiency of devices based on wurtzite NPs [43].

The scale of the NP synthesis is of high importance when considering the ink formulation because substantial gram quantities of NPs are required. Gram-scale syntheses are quite common, e.g., Houck et al. synthesized ~1 g of wurtzite CIGS NPs [28], Mousavi et al. prepared a few grams of chalcopyrite CIGS [44], and Chang et al. produced ~1.3 g of quinary Cu(In,Ga)SSe NPs with chalcopyrite structure [45]. Larger sale syntheses of CIGS NPs are less addressed, Latha et al. synthesized ~3.1 g of chalcopyrite CIGS [12], and we reported 4.5 g synthesis of CIGS NPs with chalcopyrite phase [46]. Notably, the herein reported large-scale synthesis delivers a considerable amount of ~3.5 g of wurtzite CIGS NPs.

The addition of Ga into CIS NP synthesis is known to significantly change the growth kinetics of the NPs, making the control over stoichiometry and structure challenging. Moreover, when dealing with NPs with wurtzite phase, annealing temperature of maximum 400 °C can be employed, otherwise a transformation to chalcopyrite phase will occur. In the reported synthesis, introduction of Ga gave access to phase-pure CIGS NPs with hexagonal wurtzite structure. However, in addition to being distributed inside the NPs, Ga is also segregated on the surface of the NPs. To understand this behavior, we have employed DFT calculations to estimate the relaxed atomic structures of ordered orthorhombic models, representing the bulk of wurtzite-phase CIS crystal and a thin-film slab of it. The crystallographic directions of CIS corresponding to the two open surfaces of hexagonal NPs has been identified in our previous paper [25] (Figure 6a,e). The relaxed atomic structures of these surfaces are shown in the Figure 6c,g.

Top (basal-faceted) surface of a hexagonal NP (Figure 6a–d): DFT calculations show that a monocrystalline slab with such lattice has two unequal faces: Se-rich (top) and Cu/In-rich (bottom). In the absence of passivation, while the upper Se-face is stable, the bottom Cu/In-face is less so and undergoes significant restructure to minimize tension (Figure 6c). To assess the effect of Ga dopant on CIS crystal we have repeated the calculations placing Ga instead of one of the In atoms in the bulk or on the exposed plane. The tendency of a dopant atom to segregate to the surface can be characterized through “segregation energy”: $\Delta E_{seg}=(E_{slab\left[Ga\text{-}doped\right]}-E_{slab\left[pure\right]})-\left(E_{bulk\left[Ga\text{-}doped\right]}-E_{bulk\left[pure\right]}\right)$, where $E_{bulk/slab}$ represents the total (ionic + electronic energy as calculated by DFT) of a periodic unit of a bulk system or a surface slab [47]. In case of Ga dopant in the wurtzite CIS crystal, the estimated segregation energy of ~−3 kcal mole^−1^ per dopant atom is obtained, indicating that positioning of Ga on such surface is thermodynamically favorable.

This effect is even more pronounced for the side (prism-faceted) surface (Figure 6e–h): DFT calculations show that for ideal lattice, a minimal energy surface is Cu-rich plane with the formation enthalpy (~35 kcal mole^−1^ per 144 atom super-cell) lower than a similar In-rich surface. However, surface Cu atoms are prone to substitution by Ga: we have estimated the segregation energy for a single Ga adatom from bulk to replace a surface Cu atom as −20 kcal mole^−1^. Therefore, segregation of Ga adatom to Cu-rich side surfaces of hexagonal NPs is highly favorable, leading to a self-exclusion effect of Ga in the CIGS NPs. This correlates with Ga concentration map in the Figure 2.

To assess the effect of passivating ligands, we have also conducted DFT calculations for thin-film slab surface with ethylamine group (imitating the effect of longer HDA) attached to a surface Ga, In, or Cu atoms (Figure 6d,h). By analogy, we can compare the gain from passivating either atom with ligand by calculating: $\Delta E_{passiv}=(E_{passiv\left[Ga\right]}-E_{free\left[Ga\right]})-\left(E_{passiv\left[In\right]}-E_{free\left[In\right]}\right)$. Again, we estimated $\Delta E_{passiv}$ ~−16.3 kcal mole^−1^ per Ga dopant for In-rich top surface and −28.5 kcal mole^−1^ for Cu-rich side, indicating that surface Ga atoms are energetically favorable in presence of amine group ligands. These computational results explain the Ga segregation in the **Sample I** produced in this study. The adjustment of the metal ratios and introduction of Cu vacancies in **Sample II** may provide additional stabilization factor preventing Ga segregation.

With understanding of synthesis and structural and thermal stability of synthesized CIGS NPs, a model of screen-printed photoabsorber layer was developed. Removing of the surface ligands require annealing at temperature higher than wurtzite-chalcopyrite transformation, thus the resulted film with homogenous thickness of ~4.5 µm was composed of CIGS chalcopyrite NPs. Despite the NPs’ optimal structural properties, organic residues were detected, and the resulting film presented a porous layer with low grain size. Further optimization of the film with lower porosity and better electronic properties are currently underway. In summary, the herein synthesized NPs are suitable for printing deposition of the photoabsorber layer, opening ways for the roll-to-roll production of efficient CIGS PV systems.

## 5. Conclusions

A large-scale heat-up synthesis delivered ~3.5 g of phase-pure wurtzite CIGS NPs with 10–70 nm of size. The synthesized NPs with chemical composition of Cu_0.89_In_0.72_Ga_0.28_Se_2_ have high crystallinity and wide absorption range from visible to near-infrared regions well-matching with the properties of analogous NPs used in high efficiency PV systems. Moreover, it was found that Ga, besides being distributed inside the CIGS NPs, is also segregated at the surface of the synthesized NPs. In silico calculations support that it is thermodynamically favorable for Ga atoms to segregate to the surface of wurtzite phase CIS NP both in case of non-passivated surface and in the presence of amine-based ligands. Finally, screen-printed thin films with homogenous thickness of ~4.5 µm were produced by formulating the synthesized NPs as a water-based ink, paving the way to the roll-to-roll production of CIGS PV systems.

## Figures and Tables

**Figure 1 nanomaterials-11-01148-f001:**
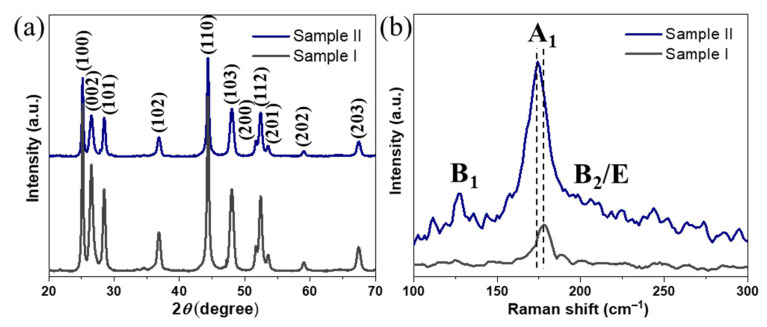
XRD patterns (ICDD card no. 01–078-5190) (**a**) and Raman spectra (**b**) of the synthesized nanoparticles from both samples.

**Figure 2 nanomaterials-11-01148-f002:**
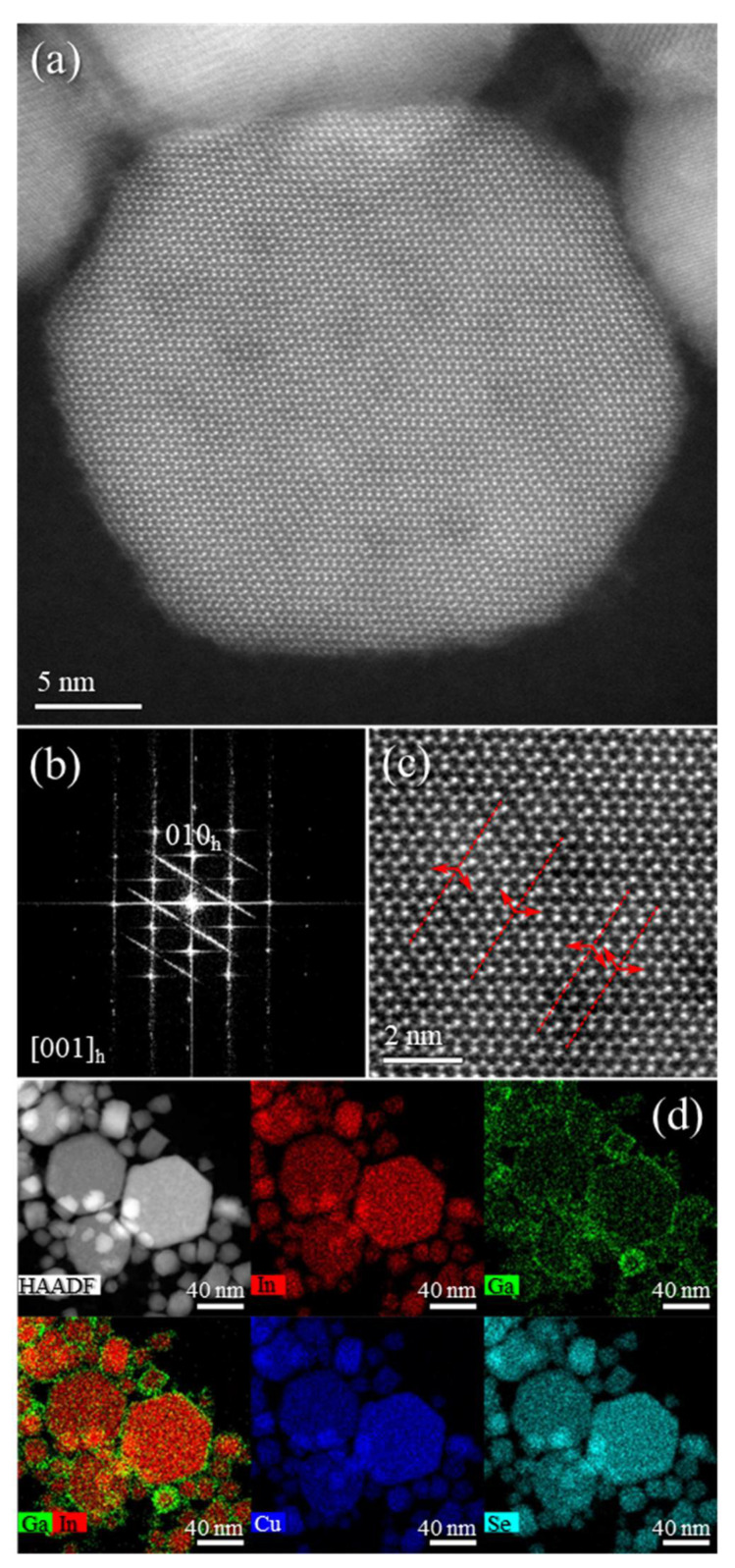
High-angle annular dark-field scanning transmission electron microscopy (HAADF–STEM) images (**a**,**c**), Fourier-transform (FFT) pattern (**b**), and STEM–energy-dispersive X-ray spectroscopy (EDX) maps (**d**) of **Sample II**.

**Figure 3 nanomaterials-11-01148-f003:**
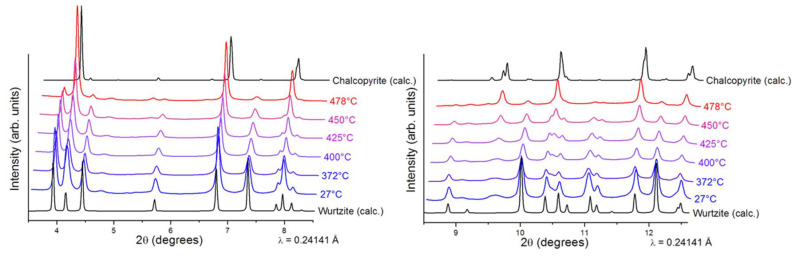
In-situ variable temperature synchrotron powder XRD of wurtzite CuInSe_2_ NPs sample (λ = 0.24141 Å). The sample was heated from room temperature (bottom pattern) to 478 °C (top pattern).

**Figure 4 nanomaterials-11-01148-f004:**
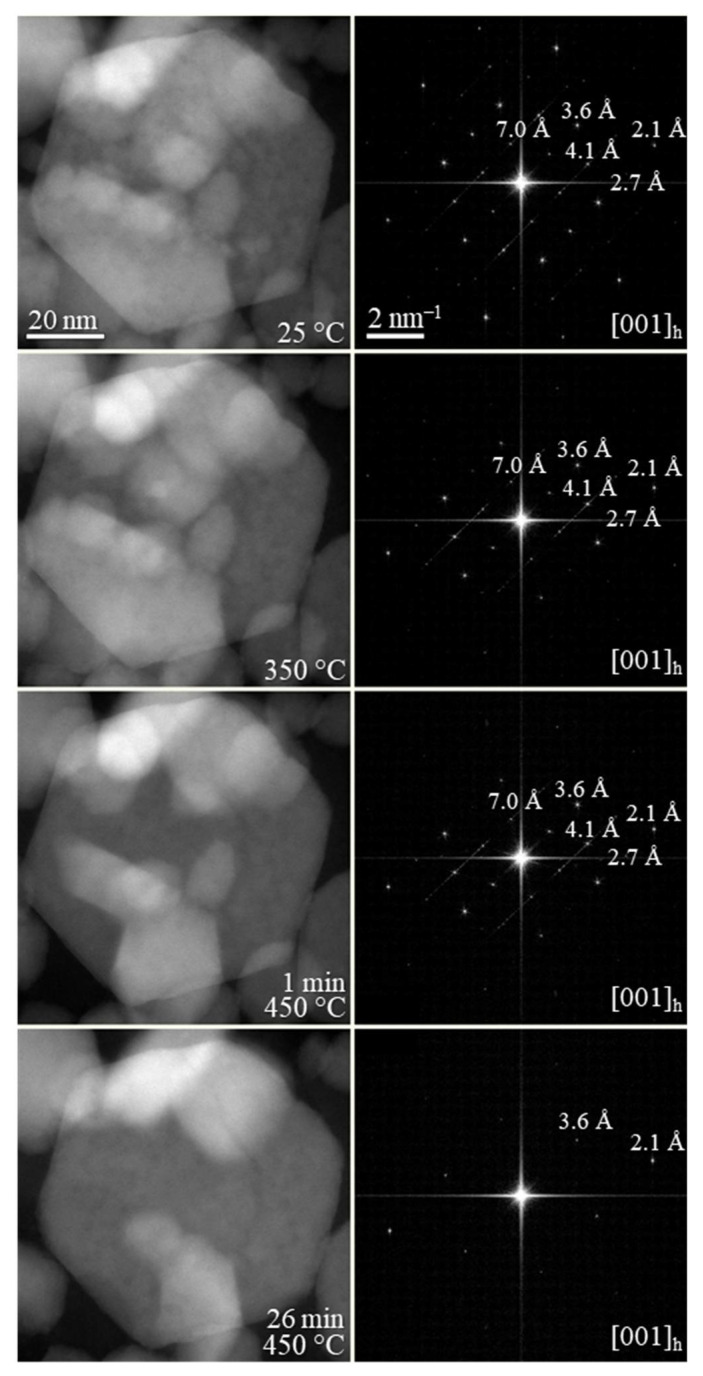
From top to bottom: in situ TEM imaging under vacuum of the synthesized Cu(In,Ga)Se_2_ NPs at RT, at 350 °C, and at 450 °C during 1 and 26 min, along with the corresponding FFT patterns.

**Figure 5 nanomaterials-11-01148-f005:**
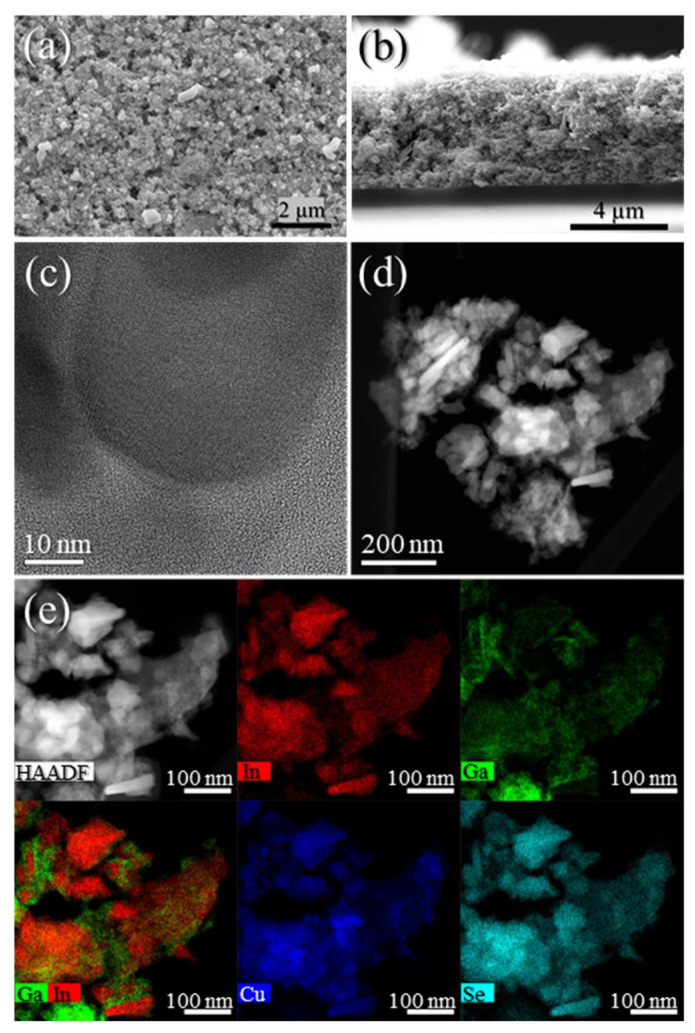
SEM surface (**a**) and cross-sectional (**b**) images and HAADF–STEM images (**c**,**d**) with STEM–EDX maps (**e**) from screen-printed film of **Sample II** after annealing.

**Figure 6 nanomaterials-11-01148-f006:**
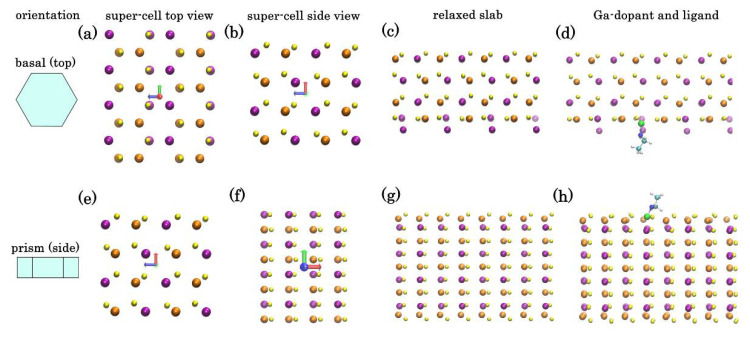
Ordered CIS orthorhombic model of the top or basal (**a**–**d**) and side or prism (**e**–**h**) surfaces of hexagonal nanoparticle: idealized super-cell top view (**a**,**e**) and side view (**b**,**f**) and density functional theory (DFT) relaxed atomic structures of the pure surface slabs (**c**,**g**) and with Ga dopant and -NH-C_2_H_5_ passivating group (**d**,**h**). Legend: In (purple), Cu (orange), Se (yellow), Ga (green), N (deep blue), C (light blue), and H (white).

## Data Availability

All data to support the conclusions of the Article are present in the Article and Appendix A. Additional data related to this article could be requested from the authors.

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
