# Peer review of "Large-Scale Synthesis of Semiconducting Cu(In,Ga)Se2 Nanoparticles for Screen Printing Application"

_nanomaterials, 2021, doi:10.3390/nano11051148_

Round 1

Reviewer 1 Report

The authors investigate the synthesis of CIGS nanoparticles and the resulting annealing films with potential application in photovoltaics. The study is well conducted, and an in-depth characterization is done. As they mentioned in the introduction, the development of photovoltaic materials is a relevant topic however, there are some key points missing in the manuscript that should be addressed:

-why the efforts to synthesize high-quality phase-pure CIGS NPs with hexagonal wurtzite phase if the resultant screen-printed film after annealing is a phase-pure CIGS with tetragonal chalcopyrite structure? Just only for stoichiometric control reasons, it is a weak reason considering the amount of work done. What about to prepare a film keeping the hexagonal wurtzite phase for PV performance? The annealing temperature can be reduced by selecting alternative ligands.

-considering the capacity of the authors demonstrated in their recent paper (https://pubs.acs.org/doi/abs/10.1021/acsaem.9b01999), why there is no photovoltaic study or at least simple photocurrent characterization details on the prepared films? This is paramount to justify the work if the final purpose is to apply the designed CIGS ink for photovoltaics.

Additionally, there are some unclear points in the work:

-why authors make a “homogenization” of the NPs using an agate mortar? If the NPs have HDA as ligand and the synthesis is well performed (stirring, heating,…) they should obtain a good size dispersion of the NPs, although images in Figure 2 show a polydisperse sample. Authors should add in the SI the statistics of the polydispersity of the samples to justify that it is necessary to improve that with the post-processing treatments they used.

-authors claim a “ligand exchange protocol” from HDA to ethylenediamine, considering that they are looking for no agglomeration, why they use a “diamine” and not another bifunctional ligand? How it is favored the ligand exchange if both ligands present the same functional group (amine)? Usually, amines can be replaced as ligands in colloidal nanocrystal dispersions using ligands containing carboxylic or thiol groups whose link with the surface is thermodynamically favored. Afterwards, they do a “wet ball milling” treatment, it is this treatment better than a sonication protocol?

-why authors made many steps for the homogenization of the NPs before arriving to the final ink? i.e. synthesis -> drying -> agate mortar -> ligand removal -> drying -> wet ball milling -> filtering -> drying -> mortar -> ink formulation. They use 3 steps for that, why not simply go from the synthesis -> drying -> ligand exchange->drying -> wet ball milling in the final ink with optional filtering? The reasons for each step should be clarified. Even more, considering that in general in photovoltaics there is a trend to go for low-cost solutions, and the increase of the number of steps is unfavorable in that sense.

-if the starting ligand (HDA) is replaced by ethylenediamine (EDA), why authors carried out the TGA analysis with the HDA-samples? According with the text, they should use the final ink (EDA-NPs) to prepare the films; or they supposed that residual HDA could remain in the NPs surface and go to evaluate the “higher boiling point” ligand? Even a TGA should be performed to ensure that the temperature used removes the HPMC added in the ink formulation. In fact, later authors commented about “residual carbon” from the HPMC.

 -in Figure S4, it is missing the comparison between the absorbance before and after the ligand exchange HDA to ethylenediamine to evaluate the success of the ligand exchange. If there is no difference in the absorption peak, then, a TGA should show the differences.

Reviewer 2 Report

Nanomaterials-1180917

Large-Scale Synthesis of Semiconducting Cu(In,Ga)Se2 Nanoparticles for Screen Printing Application

This is an interesting paper, however minor revision is needed before its publication.

1) The authors should cite more inclusively in the introduction, specifically: C. Coughlan et al. Compound Copper Chalcogenide Nanocrystals, Chem. Rev. 2017, 117, 5865, R. L. Brutchey, Diorganyl Dichalcogenides as Useful Synthons for Colloidal Semiconductor Nanocrystals, Acc. Chem. Res. 2015, 48, 2918.

2) For better characterization of this material, the Scherrer formula should be used to estimate the average crystallites size of bulk samples.

3) The authors should revise their optical band gap discussion section. In the majority of publication devoted to these inorganic semiconductors their optical band gap is estimated from (Ahv)2 vs energy (hv) relationship.

4) Page 13: „To this end, a ligand-exchange procedure was successfully employed to replace HDA, which is a solid at RT, by ethylenediamine” FTIR spectra could give additional information concerning the binding of ligands on the nanocrystals.

Round 2

Reviewer 1 Report

The authors still have not addressed the most relevant comments from the Reviewers. Here the points missing and that should be addressed:

-The authors mentioned along the text the possibility of “photovoltaic application”, and even more, in the response they comment that it is possible to use the NPs to fabricate photoabsorber layers. Therefore, authors should provide at least a simple photocurrent characterization details on the prepared films: I-V curves in dark and white light conditions. This will awake the interest of the manuscript and justify all the efforts in the ink preparation.

-If the authors now claim that “The focus of this paper is large-scale synthesis of CIGS NPs” and “most of the reported syntheses procedures give access to tetragonal chalcopyrite Cu(In,Ga)Se2 nanoparticles, whereas methods to obtain other structures are scarce”, then, why to go for a wurtzite hexagonal structure? They should justify the advantages of the wurtzite hexagonal structure vs the most common chalcopyrite phase, e.g.: more homogeneous size-distribution, reduction of annealing temperatures, or better photoresponse.

In the response letter of the authors, at the point 7, the response of the authors is uncompleted. "...as well, paragraph 6, lines 3 and ???"
